# The Impact of COVID-19 Pandemic during 2020–2021 on the Vitamin D Serum Levels in the Paediatric Population in Warsaw, Poland

**DOI:** 10.3390/nu13061990

**Published:** 2021-06-09

**Authors:** Agnieszka Rustecka, Justyna Maret, Ada Drab, Michalina Leszczyńska, Agata Tomaszewska, Agnieszka Lipińska-Opałka, Agata Będzichowska, Bolesław Kalicki, Jacek Z. Kubiak

**Affiliations:** 1Pediatric, Nephrology and Allergology Clinic, Military Institute of Medicine, 04-141 Warsaw, Poland; arustecka@wim.mil.pl (A.R.); justyna.maret@gmail.com (J.M.); adrab@wim.mil.pl (A.D.); mleszczynska@wim.mil.pl (M.L.); awawrzyniak@wim.mil.pl (A.T.); abedzichowska@wim.mil.pl (A.B.); kalicki@wim.mil.pl (B.K.); 2Cell Cycle Group, Faculty of Medicine, Institute of Genetics and Development of Rennes, University of Rennes 1, 35000 Rennes, France; jacek.kubiak@univ-rennes1.fr

**Keywords:** vitamin D, COVID-19, children

## Abstract

Background: The main source of vitamin D is skin synthesis, which depends on sunlight exposure. During the pandemic, COVID-19 children were obliged to home confinement, which potentially limiting sunlight exposure. The aim of this study was to evaluate whether home confinement led to decreased vitamin D serum levels in children in Warsaw, Poland. Methods: The study included 1472 children who were divided into two groups, based on the date of 25(OH)D level blood sampling: before and during the pandemic. Children under 1 year of age (infants) were analysed separately. Results: A statistically significant decrease in the average level of vitamin D was observed between groups of children over 1 year of age (35 ng/mL ± 18 vs. 31 ng/mL ± 14). In infants from both groups, the mean vitamin D levels were within the normal range (Group 1 inf 54 ng/mL ± 21 vs. Group 2 inf 47 ng/mL ± 15). The characteristic seasonal variability was observed before the pandemic, with maximal vitamin D levels in summer (40 ng/mL ± 17) and minimal levels in winter (30 ng/mL ± 14). During the pandemic, no seasonal variability was observed (summer 30 ng/mL ± 11 vs. winter 30 ng/mL ± 19). Conclusions: The COVID-19 pandemic restrictions led to a significant decrease in vitamin D serum levels in children.

## 1. Introduction

Optimal vitamin D levels are essential for healthy bones and the prevention and treatment of rickets [1]. Moreover, vitamin D has pleiotropic effects and regulates up to 2000 genes. It modulates the immune system and enhances immunity against infectious diseases. Vitamin D regulates the production of chemokines, prevents autoimmune inflammation, and enhances immune cell differentiation. It impacts the interaction between antigen-presenting cells and lymphocytes. It induces the production of antimicrobial peptides like cathelicidin and defensins [2]. Vitamin D deficiency was observed in the pathogenesis of type 1 diabetes mellitus, common cancers, cardiovascular diseases, multiple sclerosis, and rheumatoid arthritis [1,3,4].

Vitamin D has a limited number of sources and its availability can be easily perturbed by external factors [5]. The skin synthesis depends on sunlight exposure, thus, air pollution, living indoor, and using sunscreens lead to a decreased vitamin D production by the skin synthesis [6,7]. The other sources of vitamin D are diet and vitamin D supplementation [2]. However, the diet including fatty fish, egg yolk, and beef liver can supply only about 10% of vitamin D demand [3]. In many countries, food products fortified with vitamin D are introduced to assure optimal serum levels in the population [8]. The production and metabolism of vitamin D are illustrated in Figure 1.

In December 2019, in the Chinese region Wuhan, Hubei, a new pathogen causing the novel coronavirus disease (COVID-19) appeared. On 11 March 2020, the World Health Organization declared the COVID-19 pandemic [9]. In Poland, where we conduct our studies, the first official case of COVID-19 was reported on 4 March 2020. Due to its fast spreading and causing a high number of deaths, many countries have decided to introduce strict restrictions. New social distancing measures included home confinement and free movement limitations [10], which are likely to reduce the sunlight exposure of people.

The aim of this study was to evaluate whether home confinement has led to decreased vitamin D serum levels among the paediatric population.

## 2. Materials and Methods

This study was conducted as a retrospective analysis. It included 1472 children between the age of 1 month to 18 years. The children were admitted to the Department of Paediatrics, Paediatric Nephrology and Allergology, Military Institute of Medicine in Warsaw for diagnostics, mainly because of headache, abdominal pain, allergy. Children with metabolic disorders associated with vitamin D, calcium, and phosphate economy were excluded from this study. All children were negative for COVID-19 disease. Children were examined between January 2019 and February 2021.

The analysed data included the age of admission, serum level of 25-hydroxyvitamin D [25(OH)D], and season of blood sampling. 25(OH)D levels were examined by the electrochemiluminescence method in the Combas 6000e601 Roche analyser. Patients with 25(OH)D levels between 30–50 ng/mL were considered to have normal serum levels. Those with 25(OH)D levels lower than 20 ng/mL were considered to have vitamin D deficiency. Those between 20–30 ng/mL were considered to have vitamin D insufficiency [11].

All patients included in this study were divided into two groups, based on the date of 25(OH)D level blood sampling and called further ‘pre-pandemic’ Group 1 (January 2019 to February 2020) and ‘during pandemic’ Group 2 (March 2020 to February 2021). The mean 25(OH)D levels were compared between Group 1 and Group 2 in the overall period and the year seasons. From both groups, children under 1 year of age were distinguished and analysed separately: the infant groups were named Group 1 inf and Group 2 inf, respectively. The mean 25(OH)D levels in infants from Group 1 and Group 2 were compared separately to older patients.

The obtained data were analysed statistically using Statistica 12.0 (StatSoft, Warsaw, Poland) software. Preliminary verification of data using a distribution normality plot was performed before the analysis. Final verification was performed using the Kolmogorov–Smirnov and Lilliefors normality test. Considering the lack of compliance with normal distribution for selected variables, non-parametric tests that did not require normal distribution were used for statistical analysis in these cases. An unpaired student’s *t*-test was used to evaluate variables with normal distribution. For correlation analysis, the Spearman rank coefficient was calculated for variables without a normal distribution, and Pearson’s linear correlation coefficient was used for variables with a normal distribution. A *p*-value < 0.05 was considered statistically significant.

## 3. Results

Group 1 included 851 children (mean age 8 ± 5), among them 457 boys (54%) and 394 girls (46%). Group 2 included 621 children (mean age 8 ± 5), among them 311 boys (50%) and 310 girls (50%). Thus, no difference was noticed between the groups in terms of age and gender distribution.

The overall proportion of children with vitamin D deficiency and insufficiency recorded in Group 1 was lower than in the Group 2 (Figure 2 and Figure 3).

The mean vitamin D levels were 35 ng/mL ± 18 in Group 1 and 31 ng/mL ± 14 in Group 2 (Figure 4). A statistically significant difference in vitamin D medium serum levels between Group 1 and Group 2 was observed (Figure 4).

In children under 1 year of age, a statistically significant decrease in mean vitamin D serum levels was observed, however, a lower than in the general paediatric population analysed above (Figure 5). The mean vitamin D serum levels were 54 ng/mL ± 21 in Group 1 inf and 47 ng/mL ± 15 in the Group 2 inf. In both infant groups the mean vitamin D levels were within the normal range.

The vitamin D serum levels were also analysed in the course of different year seasons. In spring, summer, and autumn, the mean vitamin D serum levels in Group 2 were observed to be significantly lower than in Group 1. In winter, no statistically significant difference between the groups was observed (Table 1; Figure 6, Figure 7 and Figure 8).

The characteristic seasonal variability was observed in Group 1, with maximal vitamin D levels in summer and minimal levels in winter (Figure 9). In Group 2, no seasonal variability was observed (Figure 10).

## 4. Discussion

The study presented here evaluated the potential association between the pandemic-related confinement and the 25(OH)D serum levels among children. We found that the mean vitamin D levels in children decreased during the COVID-19 pandemic and increased the proportion of children with vitamin D deficiency. A similar study conducted by Yu et al., evaluated the relationship between vitamin D serum levels and pandemic-related confinement in children aged 0–6 years in Guangzhou, China. Its results were similar, showing an increase in the proportion of children with vitamin D deficiency [12].

The mean vitamin D serum levels in the group of children under 1 year of age were statistically significantly lower during the COVID-19 pandemic than before the pandemic. However, the values in both groups were within the normal range. A similar tendency was also observed by Yu et al., where the proportion of children with vitamin D deficiency decreased among children aged 0–3 years and increased among children aged 3–6 years [12]. This might be related to the vitamin D supplementation policy and more attentive care-taking of the youngest children during the COVID-19 pandemic [13].

The seasonal variability of vitamin D serum levels is commonly observed in many countries worldwide among the paediatric population [1,7]. It is mostly related to different insolation during the year. The vitamin D skin synthesis might be affected by other factors—such as the weather (hiding from heat) and the culture (lifestyle, clothing) [14,15]. The characteristic seasonal variability was observed in our research before the pandemic, with maximal vitamin D serum levels in summer and minimal levels in winter. This tendency was not observed during the pandemic. These results emphasise that the confinement could lead to reduced sunlight exposure and decreased cutaneous vitamin D synthesis.

Research conducted by Shakeri et al. showed that the vitamin D serum levels during winter months are related to vitamin D concentration at the end of summer, the age (decreasing with age), and the sex (being lower among girls) [14]. In our research, the differences in vitamin D serum levels between genders were not analysed. A tendency to lower vitamin D serum levels in girls comparing to boys was observed also in other studies [1,15]. The explanation suggested by Smyczyńska et al. is the lower physical activity of girls in outdoor activities, among them especially team sports [1]. However, in research conducted by Vierucci et al., no difference between vitamin D serum levels in girls and boys was observed [16].

Due to the COVID-19 pandemic, many negative effects on lifestyle choices were reported [17,18,19]. An online survey conducted by Sidor et al. during the nationwide lockdown in Poland indicated that the respondents were more prone to snacking (52%) and eating more (43%) [20]. Another survey on health issues conducted by Koletzko et al. reported an increase in children’s body weight (9% of children) and a decrease in physical activity (38% of all children, 60% of children aged >10 years) [17]. Cachón-Zagalaz et al. reported that only one-third of children reached the World Health Organization’s recommendations of having a minimum of 1 h per day of moderate physical exercise [21]. Moreover, Francisco et al., revealed that home confinement affected also children’s mental health [22].

Home confinement was introduced to slow down the spreading of the COVID-19 disease, but, as was shown, it reduced vitamin D serum levels. This might lead to an increased risk of respiratory infections, among them COVID-19 disease [23]. A recent study conducted by Yılmaz et al., suggested that paediatric patients diagnosed with COVID-19 disease had significantly lower vitamin D levels than those in the control group [24]. A study by Panfili et al., suggested that the supplementation of vitamin D might even play a role in the treatment of COVID-19 infection [25].

The research shows that the effects of the COVID-19 pandemic outreach the infection itself and impact the health of the paediatric population [10,17,18,19,20,21,22]. In spite of all the recommendations, supplements, and D-fortified food, the problem of vitamin D deficiency among children is still present. The importance of vitamin D in young organisms is not to be undermined. Vitamin D hypovitaminosis leads to rickets [1], to sleeping disorders [26] and was observed in the pathogenesis of type 1 diabetes mellitus [4]. Vitamin D deficiency was observed in obesity, where is associated with higher cardiovascular risk [27]. The research by Muscogiuri et al., showed that vitamin D supplementation prevents osteoporosis. Its impact on other chronic diseases still needs to be verified [28]. Another study by Muscogiuri et al., advised to consume more vitamin D-containing foods. Since the pandemic restrictions may prolong, it is recommended to have a diversified diet and pay more attention to eating habits [29].

The presented study is associated with the following limitations, which might have resulted in biased estimates. First, because of the lack of information on the eventual patients’ vitamin D supplementation and eating habits. The effect of vitamin D intake was not taken into account. Second, the exact duration of the sunlight exposure of the participants of this study was impossible to control, and thus, was not taken into account either. Third, the underlying cause of hospitalisation, as well as the parathyroid hormone serum level and the calcium and phosphate economy were not analysed. In addition, data collected in our study did not include BMI and sex.

## 5. Conclusions

The COVID-19 pandemic restrictions and home confinement led to decreased vitamin D serum levels among the paediatric population in Warsaw, Poland. The decrease was observed in all age groups, however, in children aged under 1 year, it was shown to remain within the normal range. The characteristic seasonal variability of vitamin D serum levels before the pandemic, related to alternating sunlight exposure, was not observed during the pandemic period. As adequate sunlight exposure is necessary for cutaneous vitamin D synthesis and maintaining sufficient vitamin D serum levels, our findings emphasise the importance of vitamin D supplementation in the paediatric population, especially during home confinement most probably arising from it decreased sunlight exposure.

## Figures and Tables

**Figure 1 nutrients-13-01990-f001:**
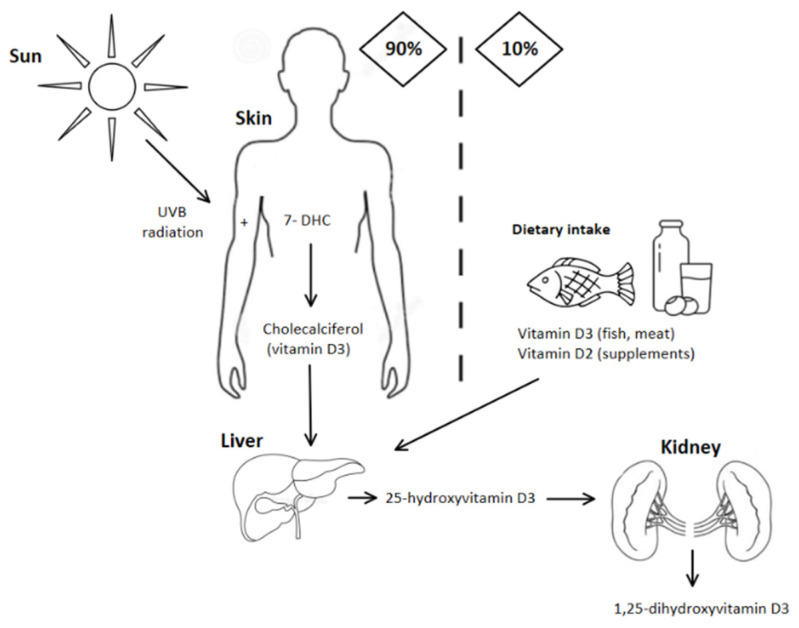
The sources and metabolism of vitamin D. About 90% of vitamin D derives from skin production. Vitamin D precursor (7-dehydrocholesterol) is converted to previtamin D. Is it further converted to vitamin D (cholecalciferol). To become the active metabolite, it is hydroxylated in the liver and then in kidneys [1,3].

**Figure 2 nutrients-13-01990-f002:**
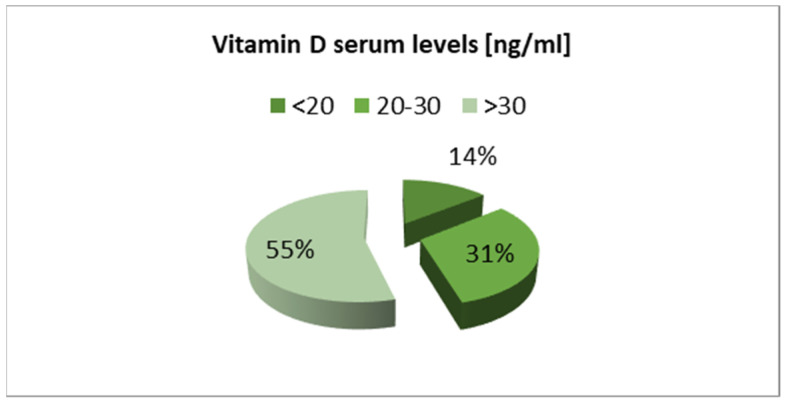
Vitamin D serum levels in children before the COVID-19 pandemic (Group 1).

**Figure 3 nutrients-13-01990-f003:**
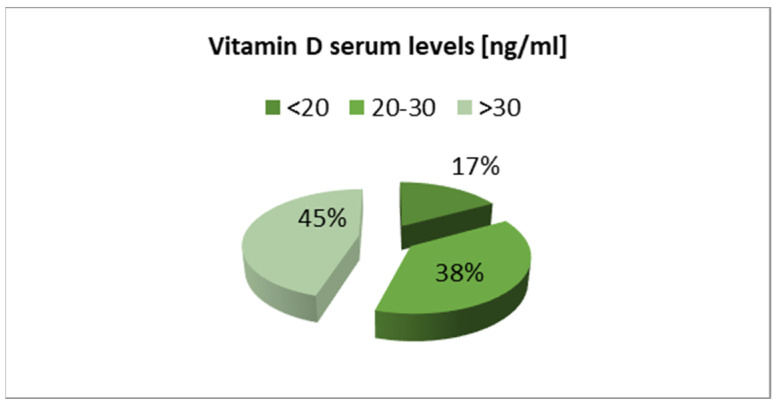
Vitamin D serum levels in children during the COVID-19 pandemic (Group 2).

**Figure 4 nutrients-13-01990-f004:**
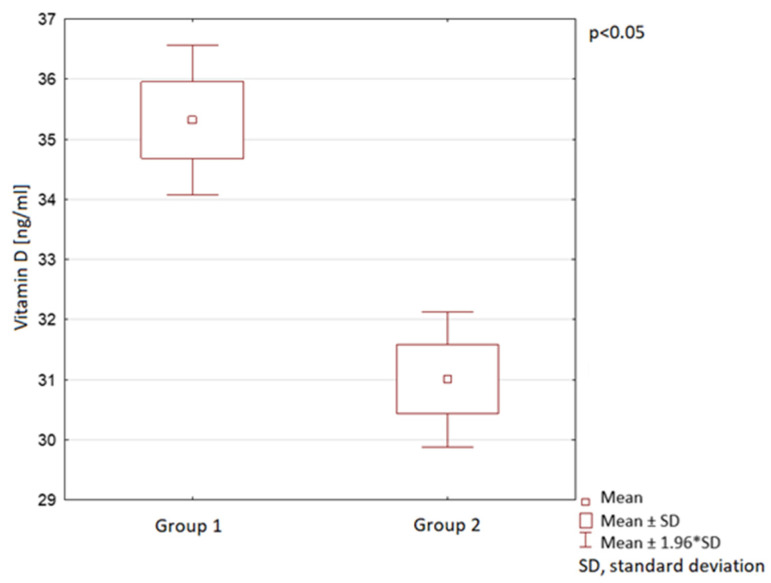
Vitamin D serum levels in children in the compared groups.

**Figure 5 nutrients-13-01990-f005:**
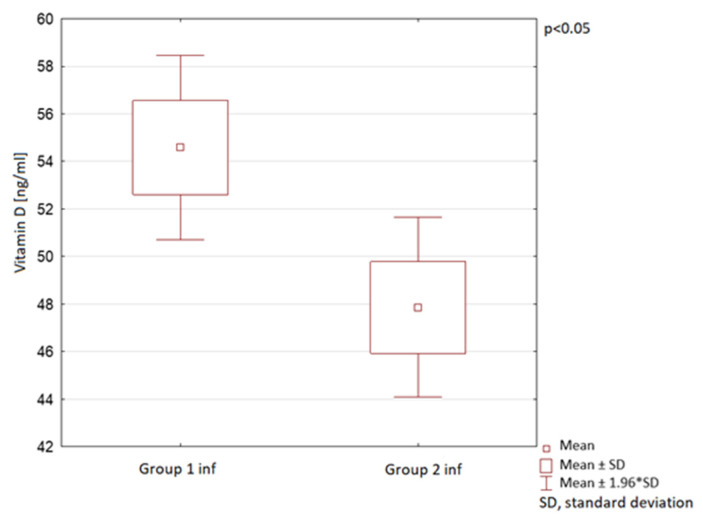
Vitamin D serum levels in children under 1 year of age in the compared groups.

**Figure 6 nutrients-13-01990-f006:**
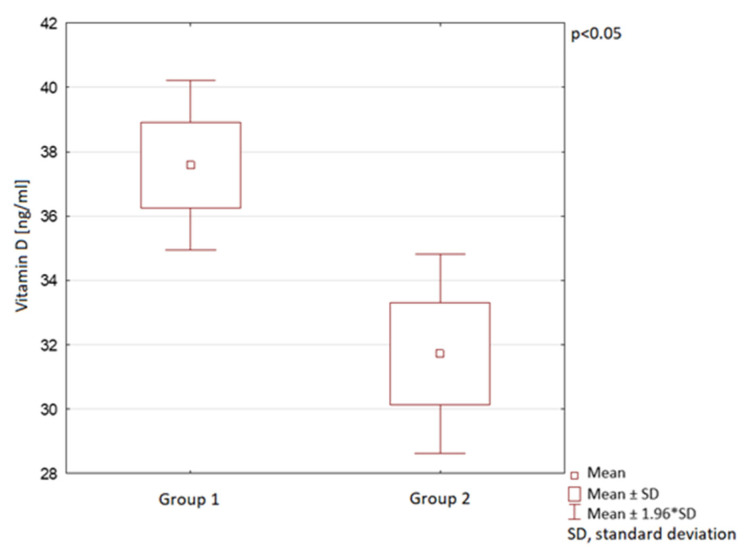
Vitamin D serum levels in spring in children in the compared groups.

**Figure 7 nutrients-13-01990-f007:**
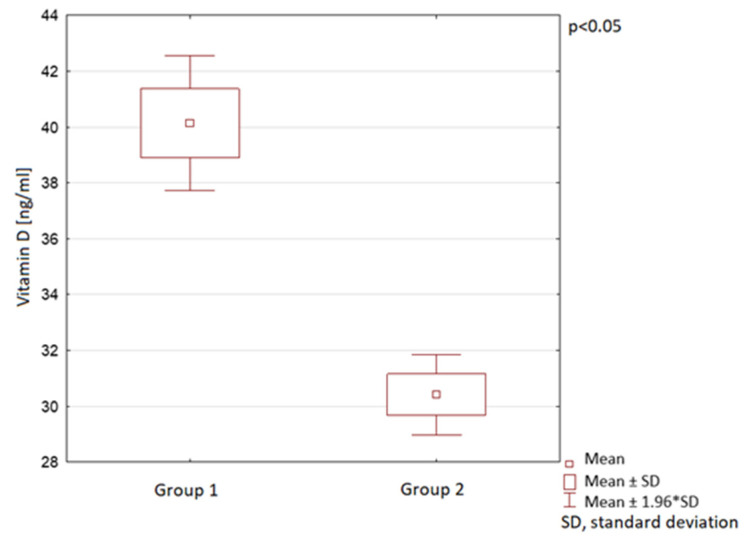
Vitamin D serum levels in summer in children in the compared groups.

**Figure 8 nutrients-13-01990-f008:**
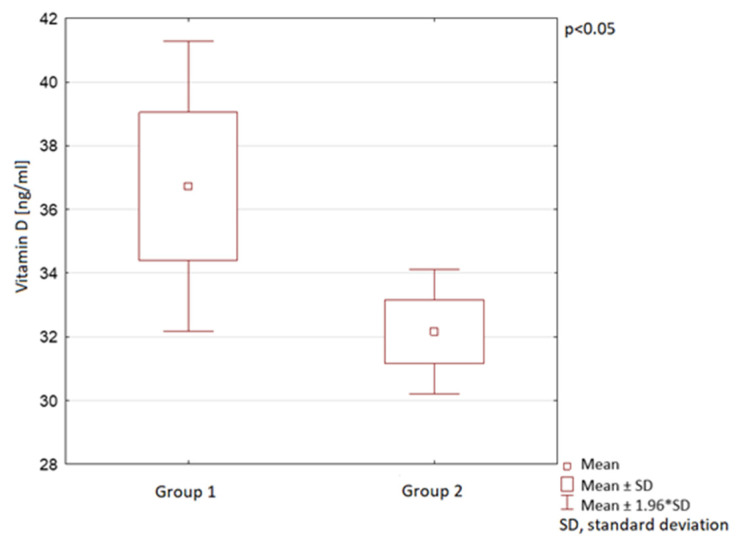
Vitamin D serum levels in autumn in children in the compared groups.

**Figure 9 nutrients-13-01990-f009:**
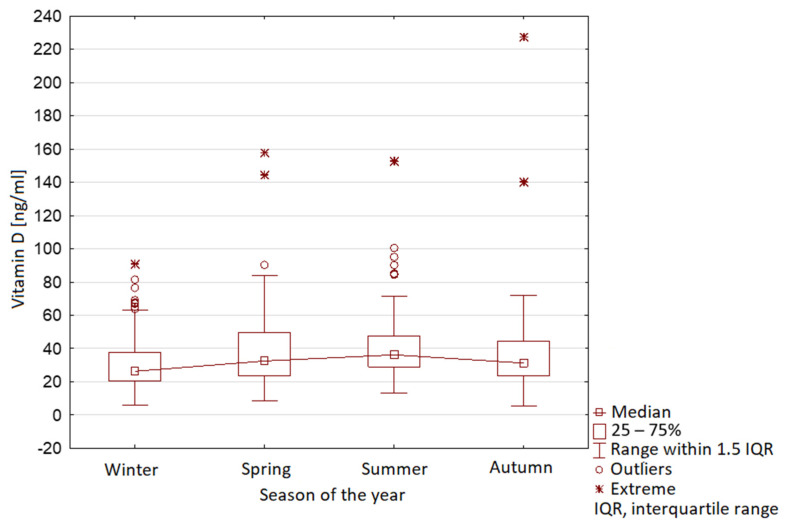
Vitamin D serum level range in year seasons in children before COVID-19 pandemic (Group 1).

**Figure 10 nutrients-13-01990-f010:**
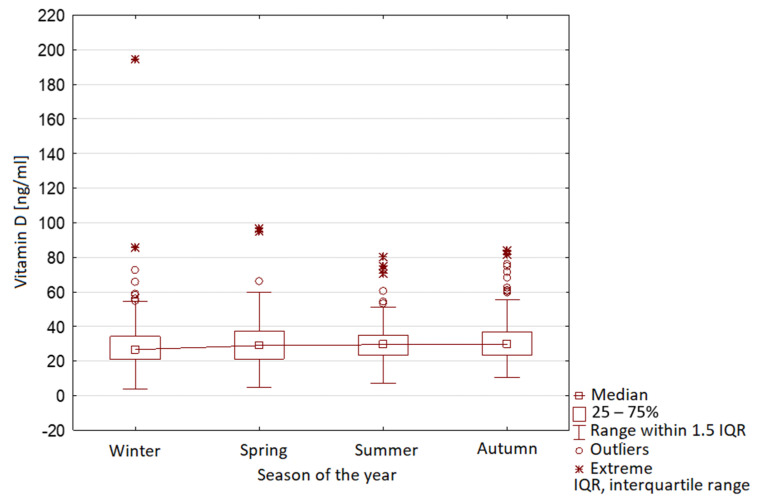
Vitamin D serum level range in year seasons in children during COVID-19 pandemic (Group 2).

**Table 1 nutrients-13-01990-t001:** Mean vitamin D serum level according to year seasons in the compared groups.

	Group 1 Vitamin D [ng/mL]	Group 2 Vitamin D [ng/mL]	*p*-Value
Spring	38 ± 20	32 ± 15	0.01 *
Summer	40 ± 17	30 ± 11	0.00 *
Autumn	37 ± 25	32 ± 13	0.04 *
Winter	30 ± 14	30 ± 19	0.87

*—statistically significant.

## Data Availability

The authors will send detailed data and calculations on request.

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
