# Peer review of "The Impact of COVID-19 Pandemic during 2020–2021 on the Vitamin D Serum Levels in the Paediatric Population in Warsaw, Poland"

_nutrients, 2021, doi:10.3390/nu13061990_

Round 1

Reviewer 1 Report

TITLE: The impact of COVID-19 pandemic during 2020-2021 on the 2 vitamin D serum levels in the paediatric population in Warsaw, 3 Poland

In the present manuscript, through a retrospective study of a large paediatric population in Warsaw, the authors identified a statistically significant decrease in serum vitamin D values as a result of the life change brought about by the COVID19 pandemic. In addition, no seasonal variation was observed during the pandemic period.

The manuscript is overall well written. Methods section is adequate to the aim, and the results are clearly presented. However, some major points deserve better attention especially in the background and discussion.

Study’s results highlight an important consequence of the lockdown imposed during this last year.

The details of the review follow:

  • It would be appropriate to make the text a little smoother, weaving evidence on the pandemic and evidence on Vitamin D through literature that studies hypovitaminosis D and the risk of Covid19 infection.

  • Line 32: the role of vitamin D within the immune system should be explained.

  • Line 36-41: it would be appropriate to replace the section explaining the synthesis of vitamin D with an equally explanatory image, underling the percentage of endogenous production by sun exposure and the percentage of production by exogenous intake.

  • It would be better to stratify the populations in the study by age groups and BMI, not only by differentiating them between children and infants
    -Sex Differences of Vitamin D Status across BMI Classes: An Observational Prospective Cohort Study (Giovanna Muscogiuri , Luigi Barrea , Carolina Di Somma , Daniela Laudisio , Ciro Salzano , Gabriella Pugliese , Giulia de Alteriis , Annamaria Colao , Silvia Savastano)

  • It would be interesting to study the eating habits of the populations concerned, especially if the change in diet during lockdown affected vitamin D values.

  • It is necessary to specify the clinical implications of this hypovitaminosis D’s pandemic.
    urr Vasc Pharmacol

-Calcium and Vitamin D Supplementation. Myths and Realities with Regard to Cardiovascular Risk (Giovanna Muscogiuri , Luigi Barrea , Barbara Altieri , Carolina Di Somma , Harjit Pal Bhattoa , Daniela Laudisio , Guillaume T Duval , Gabriella Pugliese , Cédric Annweiler , Francesco Orio , Hana Fakhouri , Silvia Savastano , Annamaria Colao)

-Vitamin D and Sleep Regulation: Is there a Role for Vitamin D?

(Fiammetta Romano , Giovanna Muscogiuri, Elea Di Benedetto, Volha V Zhukouskaya, Luigi Barrea , Silvia Savastano , Annamaria Colao , Carolina Di Somma)

-Vitamin D in obesity and obesity-related diseases: an overview

(Luigi Barrea, Evelyn Frias-Toral, Gabriella Pugliese, Eloisa Garcia-Velasquez, Maria De Los Angeles Carignano, Silvia Savastano, Annamaria Colao, Giovanna Muscogiuri)

-Vitamin D and chronic diseases: the current state of the art

(Giovanna Muscogiuri, Barbara Altieri, Cedric Annweiler , Giancarlo Balercia, H B Pal, Barbara J Boucher, John J Cannell , Carlo Foresta, Martin R Grübler, Kalliopi Kotsa, Luca Mascitelli, Winfried März, Francesco Orio, Stefan Pilz, Giacomo Tirabassi, Annamaria Colao)

  • It would be appropriate to indicate how we can stem this problem considering that the pandemic has changed our way of life almost permanently, for example citing “Nutritional recommendations for CoVID-19 quarantine” (Giovanna Muscogiuri, Luigi Barrea, Silvia Savastano, and Annamaria Colao)

Author Response

Response to Review 1

Point 1: Line 32: the role of vitamin D within the immune system should be explained.

Response 1: According to the suggestions, we have extended information on the role of vitamin D in the immune system. Line 32-36: "It modulates the immune system and enhances the immunity against infectious diseases. Vitamin D regulates production of chemokines, prevents autoimmune inflammation and enhances immune cell differentiation. Vitamin D impacts the interaction between antigen presenting cells and lymphocytes. It induces production of antimicrobial peptides like cathelicidin and defensins [2].

Point 2: Line 36-41: it would be appropriate to replace the section explaining the synthesis of vitamin D with an equally explanatory image, underling the percentage of endogenous production by sun exposure and the percentage of production by exogenous intake.

Response 2: As recommended, we have replaced the said paragraph with a graphic.

Figure 1. The sources and metabolism of vitamin D. About 90% of vitamin D derives from skin production. Vitamin D precursor (7-dehydrocholesterol) is converted to previtamin D. Is it further converted to vitamin D (cholecalciferol). To become the active metabolite it is hydroxylated in the liver and then in kidneys [1], [3].

Point 3: It would be better to stratify the populations in the study by age groups and BMI, not only by differentiating them between children and infants.

Response 3: The main aim of the study was to assess the impact of the pandemic on vitamin D concentration in children, regardless of BMI and sex. Infants were selected because vitamin D supplementation in this group is obligatory.

The data collected in our study did not include BMI and sex. However, we would consider this parameter when planning our future studies.

This informations were added as limitations of our study.

Point 4: It would be interesting to study the eating habits of the populations concerned, especially if the change in diet during lockdown affected vitamin D values.

Response 4: One of the limitations we have mentioned in our study, was the lack of information on vitamin D supplementation in patients. Unfortunately, we do not have information about patients’ diet and eating habits.

Point 5: It is necessary to specify the clinical implications of this hypovitaminosis D’s pandemic. It would be appropriate to indicate how we can stem this problem considering that the pandemic has changed our way of life almost permanently, for example citing “Nutritional recommendations for CoVID-19 quarantine” (Giovanna Muscogiuri, Luigi Barrea, Silvia Savastano, and Annamaria Colao)

Response 5: According to the suggestions, we have extended the paragraph about vitamin D defficiency and its implications in the paediatric population. Line 211-221: “The importance of vitamin D in young organisms is not to be undermined. Vitamin D hypovitaminosis leads to rickets [26], to sleeping disorders [27] and was observed in the pathogenesis of type 1 diabetes mellitus [4]. Vitamin D deficiency was observed in obesity, where is associated with higher cardio-vascular risk [28]. The research by Muscogiuri et al. showed that vitamin D supplementation prevents osteoporosis. Its impact on other chronic diseases still needs to be verified [28]. Another study by Muscogiuri et al. advised to consume more vitamin D-containing foods. Since the pandemic restrictions may prolong, it is recommended to have a diversified diet and pay more attention to eating habits [29].”

Reviewer 2 Report

An interesting original study measuring vitamin D levels in the pediatric population in Warsaw during covid 19, showing a  statistically significant reduction of mean vitamin D levels in Spring, Summer, and Autumn during the covid pandemic.

I have some queries:

In the statistical analysis subsection, please specify the maker of the program statistics and its location. Please also specify what type of t sted you used (paired, unpaired, etc...)

page 2 line 42  "Vitamin D has a limited number of sources and its availability can be easily perturbed by external factors. " this sentence needs a reference, such as: doi: 10.1007/s13668-020-00322-4.

Thank You

Author Response

Response to Review 2

Point 1: In the statistical analysis subsection, please specify the maker of the program statistics and its location. Please also specify what type of t sted you used (paired, unpaired, etc...)

Response 1: We have followed the recommendations and specified the information about the statistics. Line 92: “The obtained data were analysed statistically using Statistica 12.0 (StatSoft Poland) software.” Line 97: “An unpaired student’s t-test was used to evaluate variables with normal distribution.”

Point 2: page 2 line 42 "Vitamin D has a limited number of sources and its availability can be easily perturbed by external factors. " this sentence needs a reference, such as: doi: 10.1007/s13668-020-00322-4.

Response 2: We have added references according to the suggestion.

Reviewer 3 Report

The study by Rustecka et al. analyses the effects of confinement during pandemics on vitamin D levels.

The investigation is simple and the results expected, however it is a study which confirms the common deficiency of this hormone and supports the recommendation of its dietary supplementation, therefore it is worthy of publication. The limits of the study have been described.

I have found some minor points needing revision:

It is not clear whether some patients hospitalized during pandemics were found positive for Covid and whether their vitamin D levels were lower than the average.

References are not numbered correctly in the reference list

I suggest grouping the figures to make the comparison easier: 1 and 2, 3 and 4, 5 and 6 and 7, 8 and 9.

Author Response

Response to Review 3

Point 1: It is not clear whether some patients hospitalized during pandemics were found positive for Covid and whether their vitamin D levels were lower than the average.

Response 1: All the patients hospitalized and treated in our clinic were Covid-19-negative. Patients suspected of Covid-19 infection, who showed to emergency room, underwent PCR or antibody tests. If positive, they were transferred to infectious diseases clinic. We have added and emphasized this information in the article to make it clearer for the readers. Line 75: “All children were negative for COVID-19 disease.”

Point 2: References are not numbered correctly in the reference list. I suggest grouping the figures to make the comparison easier: 1 and 2, 3 and 4, 5 and 6 and 7, 8 and 9.

Response 2: We have corrected the references according to the suggestion.

Round 2

Reviewer 2 Report

The authors responded to all queries. The paper is publishable